Differential expression of NF-κB heterodimer RelA/p50 in human urothelial carcinoma

Durairajan Sankari 1
Jebaraj Walter Charles Emmanuel cejwalter@sriramachandra.edu.in 1
Samuel Mary Divya 2
Palani Dinesh 1
G Dicky John Davis 3
C George Priya Doss 4
Pasupati Sneha 4
Johnson Thanka 5
1 Department of Biotechnology, Sri Ramachandra Medical College and Research Institute , Chennai , India
2 Manipal Institute of Regenerative Medicine, Manipal Academy of Higher Education (formerly Manipal University) , Manipal , India
3 Department of Bioinformatics, Sri Ramachandra Medical College and Research Institute , Chennai , India
4 Department of Integrative Biology, School of Biosciences and Technology, Vellore Institute of Technology , Vellore , Tamilnadu , India
5 Department of Pathology, Sri Ramachandra Medical College and Research Institute , Chennai , India
Coates Philip
Electronic publication date: 2018 Sep 13
Publication date: 2018
Volume: 6
Electronic Location ID: e5563
Received 2018 Mar 2; Accepted 2018 Aug 11
Copyright: ©2018 Durairajan et al.
Copyright year: 2018
Copyright holder: Durairajan et al.
License: This is an open access article distributed under the terms of the Creative Commons Attribution License, which permits unrestricted use, distribution, reproduction and adaptation in any medium and for any purpose provided that it is properly attributed. For attribution, the original author(s), title, publication source (PeerJ) and either DOI or URL of the article must be cited.
License URL: https://creativecommons.org/licenses/by/4.0/

Keywords: NF-κB, RelA (p65), p50, Differential expression, Urothelial carcinoma, Diagnosis, Cytoscape, Immunohistochemistry

Funding: GATE awarded The study utilized institutional research funds, GATE awarded to Dr. Charles Emmanuel Jebaraj Walter and Dr. Dicky John Davis G. The funders had no role in study design, data collection and analysis, decision to publish, or preparation of the manuscript.

==============================
Background

Urothelial carcinoma (UC) is the fifth most common malignancy that accounts for 5% of all cancers. Diagnostic markers that predict UC progressions are inadequate. NF-κB contributes towards disease progression upon constitutive activation in many solid tumors. The nuclear localization of NF-κB indicates increased transcriptional activity while cytoplasmic localization indicates the inactive protein repository that can be utilized readily by a malignant cell. This study delineates the nuclear and cytoplasmic differential expression of NF-κB heterodimers in UC progression.

Methods

The involvement of the NF-κB proteins in UC was analyzed in silico using cytoscape. The expression of NF-κB heterodimers was analyzed by immunohistochemistry.

Results

PINA4MS app in cytoscape revealed over expression of RelA and suppression of NF-κB1 (p50 precursor) in UC whereas the expression of NF-κB target proteins remained unhindered. Immunohistochemical localization showed nuclear RelA/p50 in low grade UC whereas in high grade only RelA expression was observed. Conversely, cytoplasmic expression of RelA/p50 remained extensive across high and low grade UC tissues (p < 0.005). RelA nuclear and cytoplasmic expression (p < 0.005) was directly proportional to the disease progression. In our study, some of the high-grade UC tissues with squamous differentiation and muscle invasion had extensive nuclear p50 localization. The phenomenon of RelA/p50 expression seen increased in low-grade UC than high grade UC might be due to their interaction with other members of NF-κB family of proteins. Thus, NF-κB RelA/p50 differential expression may play a unique role in UC pathogenesis and can serve as a biomarker for diagnosis.

Introduction

In industrialized countries, urothelial carcinoma (UC) is the fifth most common cancer and second most common genitourinary tract cancer (Schulz, 2006; Lee et al., 2014). UC classification has an important role in determining treatment for patients and their prognosis (Öztürk, 2015). According to the World Health Organization grading in 2004, there are papilloma, papillary urothelial neoplasm of low malignant potential, low-grade UC and high-grade UC (Babjuk et al., 2017). The extent of tumor tissue invasion is defined by the stages Ta, T1, T2, T3 and T4. Most stages comprising Ta and T1 are composed of non-muscle invasive tumor confined to only the epithelium and lamina propria. Muscle invasive tumors invade the detrusor smooth muscle or beyond and they belong to the stages T2 to T4 (Kobayashi et al., 2014). Since bladder neoplasm is morphologically and genetically heterogeneous, it is important to differentiate these lesions as their treatment outcomes differ (Hodges et al., 2010). Low-grade UC with Ta and T1 stages require long term treatment and surveillance while high-grade T1–T4 stages would require meticulous strategizing for effective treatment inhibiting metastasis (Abol-Enein, 2008; Bellmunt et al., 2014; Miller et al., 2016). Hence, biomarkers that could predict UC progression are warranted.

Nuclear factor kappa B (NF-κB) transcription factor is an assembly of protein subunits as heterodimers that bind to a specific region on genes called the κB site for its function. Each dimer (RelA/p50; RelA/p52, RelB/p50, RelB/p52, RelA/RelB and p50/p52) has a distinct function in the cell acting upon specific stimuli. Upon induction, they act to modulate the immune system, inflammation, angiogenesis, survival and proliferation of cells (Gilmore, 2006). These effects are brought about by orchestrating more than 200 genes by the canonical pathway (classical) or the non-canonical pathway (alternate) as illustrated in Fig. 1. These important genes are tightly regulated under normal conditions, however, constitutive activation of NF-κB pathway occurs during tumorigenesis leading to aberrant expression of many target proteins (Meteoglu et al., 2008). Like other solid tumors, UC usually has a poorly understood association with an inflammatory-phenotype regulated by pro-inflammatory cytokine secretion (Masilamoni et al., 2006; Lee et al., 2012). NF-κB being the major inflammatory transcription factor, its role in UC needs exploration (Mukherjee et al., 2015). All NF-κB heterodimers act together for gene transcription while some homodimers of NF-κB are repressors of transcription. Hence we hypothesize analyzing the nuclear and cytoplasmic localization of the heterodimers in low- and high-grade UC will reveal the degree of transcriptional activation and thus the disease progression and would serve as novel biomarkers.

Figure 1 NF-κB signal transduction pathway.

Signaling cascade comprises the canonical (classical) and non-canonical (alternate) pathways involving NF-κB family proteins and their target proteins.

Material and Methods

Cytoscape analysis of NF-κB heterodimers (RelA/p50)

The NF-κB family of proteins and their representative target protein list were created by text mining of literature and from the KEGG database for NF-κB signaling pathway of Homo sapiens (04064). Uniprot IDs (provided as supplemental files, datasets) were identified for the list of proteins and analyzed in Cytoscape. This tool helps in visualization and analysis of the tissue specific expression data, RNA sequence data and interactions between/within multiple protein datasets. Protein Interaction Network Analysis for Multiple Sets (PINA4MS) app is a plug-in for cytoscape that uses protein interaction data from six public databases that are curated manually: IntAct, MINT, BioGRID, DIP, HPRD and MIPS MPact data, thus increasing the efficiency of the tool (Cowley et al., 2012; Shannon et al., 2003). The physical interactions among proteins of interest were retrieved from the Protein Interaction Network Analysis (PINA) platform, and kinase–substrate relationships were downloaded from the PhosphoSitePlus database.

Using the UniProt IDs as input we utilized the pre-existing UC tissue specific expression data set in the app to analyze the NF-κB family of proteins and their targets. Comparative analysis was performed with those proteins expressed on normal tissues and urothelial carcinoma tissues using PINA4MS app. Thresholds and limits (protein interactions only with the range 0.01–2; inter and intra spread value was set to 1 and circular cluster layout was chosen) set as default parameters of the tool were used while constructing the figure. This app uses the hypergeometric test to identify and eliminate the false discovery rate. It identifies the overrepresented terms with a correction for multiple testing using false discovery rate (the P-value < 0.05 was adjusted). The hypergeometric test was applied to test statistical enrichment of identified KEGG and reactome pathways and the P values were corrected for multiple comparisons using the Benjamini and Hochberg method. The output file of each tool was created and the functional enrichment analyses were performed on an SGE cluster using GOstats. For pathway enrichment analysis, the KEGG Orthology Based Annotation System (KOBAS) was used. The bubble diameter of the protein analyzed is representative of the degree of protein expression.

Urothelial tissues

The study was designed with 121 cases under two main groups: (i) normal tissues from subjects with no malignancy but presented with an inflammatory condition of the bladder; n = 60 and (ii) urothelial carcinoma of low grade and high grade; n = 61. The retrospective study design waived the need for obtaining informed consent from the patients, however the Institutional Ethical Committee of Sri Ramachandra Medical College and Research Institute (Deemed to be University), Chennai, India reviewed and accepted the study approving the use of FFPE tissues deposited at Sri Ramachandra Medical Centre from the year 2008 for this retrospective study (CSP/14/FEB/33/25).

Statistical analysis

Clinicopathological features were compared between the normal group and cancer group of patients. Low- and high-grades of UC were also compared to derive useful statistical associations using Mann Whitney U test and Fisher’s exact test. Pearson’s chi-square test was used for assessing differential expression of NF-κB heterodimers between the normal and cancer group. Any p value <0.05 was considered to be statistically significant. All the statistical analyses were performed using IBM SPSS v23.

Immunohistochemical analysis

Urothelial tissues fixed in 10% buffered formalin for 24 h at 4 °C were processed and embedded in paraffin as tissue blocks. FFPE tissue sections with 4 µM thickness were deparaffinized, rehydrated and incubated in Citrate buffer at pH 6.0 for 30 min under pressure for antigen retrieval. Once when the tissue sections were brought to ambient temperature, endogenous peroxidase was quenched using hydrogen peroxide (10%) for 20 min. The sections were further washed twice with tris buffered saline (TBS) at pH 7.3 and blocked with normal goat serum (Biolegend, San Diego, CA, USA) to reduce non-specific background staining. Tissue sections were incubated overnight with a rabbit polyclonal IgG anti-NF-κB p65 antibody (622602; Biolegend, San Diego, CA, USA) and with rabbit polyclonal IgG anti-NF-κB p50 antibody (sc-7178, Santa Cruz, USA) respectively, at a dilution of 1:200 each. Sections were then washed in TBS and incubated with HRP conjugated secondary antibody (goat anti rabbit IgG-HRP: sc-2004; Santa Cruz Biotechnology, Santa Cruz, CA, USA) for 2 h and visualized using 3, 3′-diaminobenzidine (Sigma Aldrich, St. Louis, MO, USA). Finally, the sections were counterstained with Mayer’s hematoxylin, dehydrated and cover-slipped. Meteoglu et al. (2008) method was followed with slight modifications. The lymphocytes in the UC tissue sections were found to be positive for the NF-κB heterodimers (RelA/p50) and hence served as positive controls. Further, normal urothelium devoid of blood vessels served as negative control (No primary antibody, reagent control).

The immunohistochemical staining was documented using Olympus BX43 microscope with a QImaging Micropublisher 3.3 RTV camera and QCapture Pro 7 software (QImaging, Surrey, BC, Canada). Each section was assessed for the immunoperoxidase staining and scored by a pathologist. The scoring includes RelA and p50 nuclear and cytoplasmic staining positivity. RelA and p50 staining in the nucleus and cytoplasm were scored as follows (i) absence of staining (no or non-specific), (ii) low (<20%), (iii) moderate (21–60%) and (iv) extensive (>60%) staining.

Results

In silico differential expression of NF-κB family of proteins and their targets in UC and normal urothelium

Cytoscape app PINA4MS extracted the differentially expressed proteins of normal and UC tissues. NF-κB family proteins (marked in red) like RelA, IKBKB (IKKβ), NF-κBIE (IκBε), IKBKG (NEMO), CHUK (IKKα) were over expressed in UC when compared to normal urothelial cells. However, expressions of other NF-κB family proteins like BCL3, NF-κBIA (IκBα), NF-κBIB (IκBβ), RelB, NF-κB1 (p105) and NF-κB2 (p100) were suppressed as depicted by the diameter of the bubble (Figs. 2 and 3). The NF-κB target proteins (marked in green), like CFLAR (c-FLIP), TRAF2, XIAP, MYC, BIRC3, TRAF1, BCL2, CCND1, IL8, ICAM1 and VCAM1 showed altered expression (Figs. 2 and 3). These proteins are either direct or indirect targets of NF-κB signal transduction. Of the target proteins, VEGF-A, SOD2, and PTGS2 (COX2) showed over expression, however, TRAF2, CFLAR (c-FLIP), XIAP, BIRC3 (cIAP-2), BIRC2 (cIAP-1), MYC and BAD proteins showed homogenous expressions while CCL4 (MIP-1β), CCND1 (Cyclin D1), ICAM1, BCL2, MMP9 and TRAF9 proteins showed a more suppressed state (Figs. 2 and 3). The app resulted in a direct association between RelA and NF-κB1 (p50 precursor) with a difference in their expression levels between UC and normal tissues. Hence, the localization of RelA/ p50 heterodimers was further analyzed using immunohistochemistry in UC and normal urothelium.

Figure 2 Cytoscape-PINA4MS result of NF-κB family proteins expression and targets in urothelial carcinoma.

NF-κB proteins (red bubbles) and their target proteins (green bubbles) expressed or repressed in UC tissues. Size of the bubble represents the expression at mentioned condition. Transparency of the bubble is proportional to the secretion and subsequent expression or suppression. Purple arrows indicate protein–protein interactions whereas magenta arrows indicate the substrate-kinase interactions. All interactions in UC are marked red irrespective of their interaction mode. Proteins with no arrows (stand alone) indicate indirect interaction with the members of the network. Proteins with line fill indicate those with no human protein atlas data.

Figure 3 Cytoscape-PINA4MS result of NF-κB family proteins expression and targets in Normal Urothelium.

NF-κB proteins (red bubbles) and their target proteins (green bubbles) expressed in normal urothelial tissues. Size of the bubble represents the expression at mentioned condition. Transparency of the bubble is proportional to the secretion and subsequent expression or suppression. Purple arrows indicate protein–protein interactions whereas magenta arrows indicate the substrate-kinase interactions. All interactions in UC are marked red irrespective of their interaction mode. Proteins with no arrows (stand alone) indicate indirect interaction with the members of the network. Proteins with line fill indicate those with no human protein atlas data.

Clinicopathological correlations with NF-κB heterodimer expression in UC and normal urothelium

The clinicopathological parameters like age, gender, tumor size and grades of UC were assessed to find the significance between nuclear and cytoplasmic localization of NF-κB heterodimers. The age of patients in the normal group ranged from 20 to 79 years and those in the cancer group ranged from 18 to 88 years. It was a heterogeneous population consisting of 32 males and 28 females in the normal group whereas 49 males and 12 females in cancer group. UC occurrence was higher in the age category of 61 to 70 years and found to have a high incidence in the male population with a statistical significance of p < 0.05. Comparison of tumor size, grades had high significance (p < 0.005) and revealed significant association with RelA nuclear (p < 0.005) and cytoplasmic localization (p < 0.005). However, there was no association with p50 nuclear localization (p > 0.05) but there was a significant association with the cytoplasmic expression (p < 0.05). Those cases that had muscle invasive areas showed differential uptake of the NF-κB subunits irrespective of grade.

The nuclear expression of RelA when compared between normal and UC groups by Pearson’s chi-square test (χ2) revealed a strong association with UC and nuclear localization (Table 1). Comparison of the high and low grades of UC by Fisher’s exact test showed no association with RelA nuclear localization and UC grades (Table 2). The cytoplasmic expression of RelA was compared with normal and UC group by Pearson’s chi-square test and it revealed a significant association with UC (Table 3). Comparing high and low grades of UC and the cytoplasmic localization of RelA by Fisher’s exact test revealed significant associations for RelA cytoplasmic localization (Table 4). Similarly, the nuclear expression of p50 was compared between normal and UC groups by Pearson’s chi-square test which had significant associations (Table 1) with UC and nuclear localization. Comparison of high and low grades of UC by Fisher’s exact test revealed no significant associations with p50 nuclear localization and UC grades (Table 2). When cytoplasmic expression of p50 was compared in normal and UC group by Pearson’s chi-square test it exhibited high statistical significance in UC (Table 3). Comparing the cytoplasmic expression of p50 within high and low grades of UC using Fisher’s exact test, showed significant association for p50 cytoplasmic localization (Table 4). The normal urothelium had no significant associations in the localization of the NF-κB heterodimers. The inflammatory regions had weak to moderately stained heterodimers in the cytoplasm and the nucleus had only negligible staining, unlike neoplastic cells (Tables 1 and 3).

Table 1 Nuclear localization of NF-κB heterodimers in urothelial tissues: n (%).

Tissue		NF-κB heterodimers	χ2 test/Fisher’s test	p value	
		Absent	Present			
Normal	RelA	60 (100%)	0 (0%)	39.236	<0.001	
Cancer	31 (50.8%)	30 (49.2%)	
Normal	p50	60 (100%)	0 (0%)	9.564	<0.005	
Cancer	52 (85.2%)	9 (14.8%)	

Table 2 Nuclear localization of NF-κB heterodimers in urothelial carcinoma tissues: n (%).

Cancer grade		NF-κB heterodimers	χ2test/Fisher’s test	p value	
		Absent	Present			
Low	RelA	18 (64.3%)	10 (35.7%)	3.755	0.073	
High	13 (39.4%)	20 (60.6%)	
Low	p50	24 (85.7%)	4 (14.3%)	0.009	1.000	
High	28 (84.8%)	5 (15.2%)	

Table 3 Cytoplasmic localization of NF-κB heterodimers in urothelial tissues: n (%).

Tissue		I	II	III	χ2test/Fisher’s test	p value	
Normal	RelA	24 (40%)	36 (60%)	0 (0%)	67.944	<0.001	
Cancer	0 (0%)	21 (34.4%)	40 (65.6%)	
Normal	p50	24 (40%)	36 (60%)	0 (0%)	78.644	<0.001	
Cancer	0 (0%)	15 (24.6%)	46 (75.4%)	
Notes.

Cytoplasmic positivity, I, low (≤20%); II, moderate (21–60%); III, extensive (≥61%) was based on the staining intensity of the cell cytoplasm with immunoperoxidase positivity.

Table 4 Cytoplasmic localization of NF-κB heterodimers in urothelial carcinoma tissues: n (%).

Cancer grade		II	III	χ2 test/Fisher’s test	p value	
Low	RelA	14 (50%)	14 (50%)	5.561	0.030	
High	7 (21.2%)	26 (78.8%)	
Low	p50	11 (39.3%)	17 (60.7%)	6.028	0.019	
High	4 (12.1%)	29 (87.9%)	
Notes.

Cytoplasmic positivity, I, low (≤20%); II, moderate (21–60%); III, extensive (≥61%) was based on the staining intensity of the cell cytoplasm with immunoperoxidase positivity.

There were no cases in the “low cytoplasmic positivity (I)” category.

Differential expression of NF-κB subunits in high grade UC

The expression of RelA and p50 are described based on the grade and stage of UC for clarity. H&E staining revealed histology of a non-invasive high grade UC with Ta stage (Fig. 4A). RelA expression in high grade UC at Ta stage had most of the urothelial cells positive for nuclear expression along with extensive cytoplasmic expression (Fig. 4B). However, nuclear staining was absent for p50 in high grade UC at Ta stage but all urothelial cells had moderate to extensive cytoplasmic expression (Fig. 4C). The focal areas of squamous cell differentiation in high grade UC (5%) also showed nuclear positivity (Fig. 4D). RelA nuclear expression in high grade UC at T1 stage was low whereas its cytoplasmic expression remained extensive (Fig. 4E). In high grade UC at T1 stage, p50 expression was moderately positive in the nucleus with extensive cytoplasmic expression (Fig. 4F). H&E staining revealed the histology of high grade invasive UC with focal squamous differentiation and muscle invasion at T2b stage (Fig. 4G). Nuclear RelA expression in high-grade UC at T2 stage was low and the cytoplasmic expression was moderate to extensive (Fig. 4H). Nuclear p50 expression was low but had extensive cytoplasmic staining in T2 stage of UC. Muscle invasive areas and regions with squamous differentiation had intense staining of p50 subunit (Fig. 4I). High-grade UC at T3 stage showed moderate expression of RelA in the nucleus and cytoplasm. However, p50 nuclear staining was absent and showed moderate cytoplasmic positivity.

Figure 4 NF-κB heterodimer expression: High grade urothelial carcinoma.

(A) High grade non invasive urothelial carcinoma H&E × 200. (B) Immunostaining with RelA (p65) showed nuclear positivity (brown colour) indicated by arrow IHC × 200. (C) NF-κB p50 showed moderate cytoplasmic positivity IHC × 200. Note: adjacent lymphocytes (in-built control) have taken up nuclear staining indicated by arrows. (D) High grade papillary invasive urothelial carcinoma with squamous differentiation H&E × 200. (E) Immunostaining with RelA (p65) showed nuclear positivity (brown colour) as indicated by arrows IHC × 200. (F) NF-κB p50 showed few nuclear and cytoplasmic positivity in squamous regions which are indicated by arrows IHC × 200. (G) High grade invasive urothelial carcinoma H&E × 100. (H) Immunostaining with RelA (p65) showed nuclear positivity (brown colour) as indicated by arrows IHC × 200. (I) NF-κB p50 showed extensive cytoplasmic positivity as indicated by arrows IHC × 200. Scale bar indicates 50 µM.

NF-κB heterodimer expression varies in low-grade UC and normal urothelium

H&E staining revealed the histology of low-grade papillary UC at Ta stage (Fig. 5A). The expression of NF-κB heterodimers was prominent with extensive nuclear and cytoplasmic expression of RelA (Fig. 5B). However, nuclear expression of p50 in low-grade UC at Ta stage was observed along with extensive cytoplasmic expression (Fig. 5C). H&E staining revealed tumor histology suggestive of low-grade papillary UC with T1 stage (Fig. 5D). Also, low-grade papillary UC at T1 stage had extensive nuclear staining of RelA with equivalent expression in the cytoplasm (Fig. 5E). Nuclear expression of p50 in low-grade UC at T1 stage was absent but it expressed moderate to extensive cytoplasmic staining (Fig. 5F). RelA/p50 subunits had negligible nuclear positivity and low to moderate cytoplasmic positivity in normal urothelial tissues. The regions of inflammation had some prominent nuclear positive cells of RelA/p50 subunits but they did not show any significance. However, the cytoplasm showed a difference in the staining intensity for both subunits when compared to their nuclear staining which was restricted to cells with inflammation (Figs. 5H, 5I).

Figure 5 NF-κB heterodimer expression: low grade urothelial carcinoma with normal urothelium.

(A) Low grade papillary urothelial carcinoma H&E ×100. (B) Immunostaining with RelA (p65) showed nuclear positivity (brown colour) IHC × 100. (C) NF-κB p50 showed extensive cytoplasmic positivity IHC × 100. (D) Low grade papillary urothelial carcinoma with tumor infiltration H&E × 40. (E) Immunostaining with RelA (p65) showed nuclear positivity (brown colour) IHC ×200. (F) NF-κB p50 showed moderate cytoplasmic positivity IHC × 200. (G) Normal urothelium H&E × 200. (H) Immunostaining with RelA (p65) showed faint to moderate cytoplasmic positivity (brown colour) IHC × 200. Note: Lymphocytes (in-built control) have taken up stain indicated by arrows. (I) NF-κB p50 showed moderate cytoplasmic positivity indicated by arrow IHC × 200. Scale bar indicates 50 µM.

Discussion

The role of NF-κB pathway proteins in the development and progression of UC is still unexplored. Constitutively active NF-κB tends to protect the cells from apoptosis and cross-talks with other pathways that are capable of retaining proliferating cells viable during the process of carcinogenesis (Kim et al., 2013; Mukherjee et al., 2017). Although NF-κB heterodimers act together for gene activation, homodimers of p50 repress transcription by binding to DNA. Furthermore, there are other binding partners for both the subunits to elicit their function. The degree of transcriptional activation which denotes UC progression can be understood from the variations observed in their nuclear/cytoplasmic expression. NF-κB nuclear localization indicates increased transcriptional activity and cytoplasmic localization indicates the inactive protein repository that can be utilized readily by a malignant cell. Differentiating these neoplasms is crucial as they can influence treatment outcomes (Hodges et al., 2010). This study analyzing the nuclear and cytoplasmic localization of the heterodimers in low and high grade UC unravels the disease progression.

In silico analysis was performed to assess the expression of NF-κB family of proteins and their target proteins using the cytoscape app PINA4MS. The app virtually assessed the tissue specific protein expression utilizing the data deposited in six different databases. A list of NF-κB family of proteins with their target proteins used as inputs showed their expression in normal and UC. Our results revealed suppression of NF-κB1 (p50 precursor), NF-κB2 (p52 precursor) and RelB in urothelial tumor tissue (Fig. 2). We also found a reduction in the expression of CHUK (IKKα) and similar regulatory proteins of NF-κB in the tumor cells in comparison to normal (Fig. 3). NF-κBIE (IκBε) and IκBKB (IKKβ) are the upstream molecules that showed overexpression in tumor tissue when compared to normal urothelial tissue. These results convey that the dysregulated upstream molecules of the NF-κB contribute to the constitutive activation of the NF-κB pathway resulting in an over expression of NF-κB downstream target proteins to enrich the tumor microenvironment convincing earlier studies of (Slattery et al., 2018). Accordingly, most of the NF-κB direct target proteins like XIAP, BCL3, TRAF2, BIRC3, RelB, SOD2, CFLAR or the indirect target proteins like MYC, BAD, PTGS2/COX2, CxCL2, VEGFA, VCAM1, ICAM1 found to be expressed in normal urothelium where seen selectively expressed or suppressed in the case of UC (Fig. 2). Our study revealed over expression of pro-survival or pro-proliferative factors (TRAF-1, TRAF-2, CFLAR, XIAP, BIRC2, BIRC3, VEGF-A, BAFF) and suppression of NF-κB inhibitory factors (NF-κB1E, NF-κB1, NF-κB1A, NF-κB1B) in UC.

PINA4MS app in cytoscape revealed involvement of NF-κB direct or indirect target proteins that contribute to UC carcinogenesis by modulating the steps of carcinogenesis (oncogenic control, cellular transformation, proliferation, invasion, angiogenesis and metastasis) as illustrated in Fig. 1. The expression of NF-κB target proteins remained unhindered whereas over expression of RelA and suppression of NF-κB1 (p50 precursor) was observed in UC which was further analyzed by immunohistochemistry.

Oya et al. (2003) state that affirming NF-κB activity with only RelA expression would be insufficient to conclude a finding since the binding of target genes is brought about by both subunits. However, many cancer studies have assessed the involvement of the RelA subunit alone or in combination with other subunits like RelB, NF-κB2, c-Rel, etc. Nonetheless, outcome of the studies that methodically examined RelA and p50 have reported altered expression with an impact on recurrence and survival (Jenkins et al., 2007; Annunziata et al., 2010; Wu et al., 2015). Our study has analyzed the expression of NF-κB subunits, RelA/p50 using immunohistochemistry hypothesizing a difference in their expression as supported by the preliminary in silico results. The clinicopathological parameters such as age, gender, tumor size, stage and grade were assessed with the localization of the RelA/p50 heterodimers. Geriatric males had the highest incidence of UC which can be attributed to an occupational hazard and/or habit and/or lifestyle which was not assessed in this study. The association of tumor size with nuclear localization had significance with RelA (p < 0.05) but not with p50 subunit (p = 0.655).

Our observation on the expression of RelA subunit in UC was similar to the pattern reported by Levidou & Saetta (2008) while the clinical significance of our study was the expression of NF-κB p50 subunit. The heterodimers are retained in the cytoplasm in their inactive form which upon activation translocates to the nucleus for gene transcription. At a given instance, not all the inactive heterodimers are capable of getting activated upon stimulation for translocation since this step is tightly regulated as shown by (King et al., 2011). However, the amount of inactive heterodimers in the cytoplasm shows the capacity of the cell to keep the signaling cascade active. This reflects the potential of over expressed NF-κB to manipulate UC aggressiveness and or progression, hence considering its expression seemed rational. Accordingly, we observed moderate to extensive cytoplasmic localization of RelA and p50 in all low and high grades of UC (Ta, T1, T2 and T3) investigated (Fig. 4). RelA was seen localized in the nucleus of all high grade stages of UC but only a few cases localized p50 (specifically T1, T2) which portrays the differential expression of NF-κB heterodimers in UC. A similar pattern was observed in renal cell carcinoma (Sweeney et al., 2004) where the expression of p50 in the nucleus was seen decreased. The nuclear expression of RelA and p50 in low grade Ta was positive and their expression was comparable though the p50 expression was not extensive like RelA (Tables 1 and 2). RelA was seen localized in the nucleus of all low grade, stages (Ta, T1) of UC conversely NF-κB p50 subunit was seen localized only in Ta stage UC (Fig. 5). Our study revealed that those tissues with muscle invasion and squamous differentiation had altered expression of p50. In intraepithelial neoplasia, a precancerous condition of the prostate also showed increased p50 subunit with other binding partners like RelB and p52 over the classic heterodimer expression supporting disease progression (Lessard et al., 2006; Degoricija et al., 2014). In our study, we have not assessed the expression of RelB or p52 but the pattern of RelA/p50 heterodimer expression studied was similar to Lessard et al. (2006) as reported in prostate cancer. The increased nuclear localization seen in the low grade UC than high-grade UC is suggestive of the contribution of NF-κB heterodimer in the initial stages of UC pathogenesis.

Nuclear expression of p50 homodimers can be independent of RelA since they can form complexes with other proteins like CBP, Bcl-3, histone deacetylase 1 (HDAC-1) and p300 for DNA binding to initiate or repress gene transcription based on the binding partner (Karst et al., 2009). Hence the involvement of p50 in UC is perhaps independent of the I κB regulation and or in combination with other transcriptional activators for gene transcription. In normal conditions, the homodimers of p50 are reported to be repressors and they are up-regulated for inhibiting the NF-κB signaling (Annunziata et al., 2010). In UC, we speculate that this inhibitory mechanism is dysregulated. In our study lymphocytes served as in-built controls of RelA/p50 staining (Figs. 4C, 4H). Tumor-associated macrophages are known to show nuclear p50 positivity attributing to anti-tumor response suppression (Saccani et al., 2006; Annunziata et al., 2010) and many investigations state the presence of tumor infiltrative macrophages with poor prognosis (Szebeni et al., 2017). The significance of lymphocytes showing NF-κB heterodimers positive staining in UC is unknown. Muscle invasion and squamous differentiation are indicators of poor prognosis with adverse outcome in UC (Amin, 2009; Lee et al., 2014). Selective nuclear localization of p50 in certain regions of UC tissues in our study warrants further investigation on the actual mechanism of action in UC pathogenesis. Annunziata et al. (2010) have associated NF-κB p50 with poor survival in ovarian cancer patients post-treatment. Similarly, this study on UC that demonstrates differential p50 expression is suggestive of its participation in disease progression, and hence targeting it would help patient’s early diagnosis and as a novel biomarker.

Conclusion

Differential nuclear expression of the NF-κB subunits RelA/p50 poses important questions on their functions in UC. RelA must dimerize for DNA binding activity as they lack the DNA binding domain. From this study, it becomes evident that RelA has a partner that allows it to transcribe the target genes. The lack of extensive nuclear p50 suggests that RelA translocates with another binding partner instead of p50. Among the NF-κB family of proteins, identifying the RelA preferred binding partner during UC progression will shed some light on the role of RelA in the progression of UC. In addition, preventing the migration of RelA by blocking the interaction of its preferred nuclear partner may abrogate the progression of UC. Investigating the significant cytoplasmic expression of p50 similar to RelA across UC tissues when it is not the preferred binding partner will also add some interesting facts to the NF-κB mystery in UC. Altogether, this phenomenon could be contemplated as the capacity of the neoplastic cell to activate the over expressed cytoplasmic NF-κB constitutively for tumor promotion, addressing these critical questions would help in early diagnosis, aid in better treatment and management of UC.

Supplemental Information

Supplemental Information 1 Proteins of the NFKB family along with respective UniProt IDs

Click here for additional data file.

Supplemental Information 2 Targets of the NFKB family proteins as retrieved from KEGG pathway along with its respective UniProt IDs

Click here for additional data file.

Supplemental Information 3 NF-κB heterodimer expression: High grade urothelial carcinoma

(A) High grade non invasive urothelial carcinoma H&E × 200.

Click here for additional data file.

Supplemental Information 4 NF-κB heterodimer expression: High grade urothelial carcinoma

(B) Immunostaining with RelA (p65) showed nuclear positivity (Brown colour) indicated by arrow IHC × 200.

Click here for additional data file.

Supplemental Information 5 NF-κB heterodimer expression: High grade urothelial carcinoma

(C) NF-κB p50 showed moderate cytoplasmic positivity IHC × 200. Note: adjacent lymphocytes (in-built control) have taken up nuclear staining indicated by arrows.

Click here for additional data file.

Supplemental Information 6 NF-κB heterodimer expression: High grade urothelial carcinoma

(D) High grade papillary invasive urothelial carcinoma with squamous differentiation H&E × 200.

Click here for additional data file.

Supplemental Information 7 NF-κB heterodimer expression: High grade urothelial carcinoma

(E) Immunostaining with RelA (p65) showed nuclear positivity (Brown colour) as indicated by arrows IHC × 200.

Click here for additional data file.

Supplemental Information 8 NF-κB heterodimer expression: High grade urothelial carcinoma

(F) NF-κBp50 showed few nuclear and cytoplasmic positivity in squamous regions which are indicated by arrows IHC × 200.

Click here for additional data file.

Supplemental Information 9 NF-κB heterodimer expression: High grade urothelial carcinoma

(G) High grade invasive urothelial carcinoma H&E × 100.

Click here for additional data file.

Supplemental Information 10 NF-κB heterodimer expression: High grade urothelial carcinoma

(H) Immunostaining with RelA (p65) showed nuclear positivity (Brown colour) as indicated by arrows IHC × 200.

Click here for additional data file.

Supplemental Information 11 NF-κB heterodimer expression: High grade urothelial carcinoma

(I) NF-κBp50 showed extensive cytoplasmic positivity as indicated by arrows IHC × 200.

Click here for additional data file.

Supplemental Information 12 NF-κB heterodimer expression: Low grade urothelial carcinoma with normal urothelium

(A) Low grade papillary urothelial carcinoma H&E × 100.

Click here for additional data file.

Supplemental Information 13 NF-κB heterodimer expression: Low grade urothelial carcinoma with normal urothelium

(B) Immunostaining with RelA (p65) showed nuclear positivity (Brown colour) IHC × 100.

Click here for additional data file.

Supplemental Information 14 NF-κB heterodimer expression: Low grade urothelial carcinoma with normal urothelium

(C) NF-κBp50 showed extensive cytoplasmic positivity IHC × 100.

Click here for additional data file.

Supplemental Information 15 NF-κB heterodimer expression: Low grade urothelial carcinoma with normal urothelium

(D) Low grade papillary urothelial carcinoma with tumor infiltration H&E × 40.

Click here for additional data file.

Supplemental Information 16 NF-κB heterodimer expression: Low grade urothelial carcinoma with normal urothelium

(E) Immunostaining with RelA (p65) showed nuclear positivity (Brown colour) IHC × 200.

Click here for additional data file.

Supplemental Information 17 NF-κB heterodimer expression: Low grade urothelial carcinoma with normal urothelium

(F) NF-κBp50 showed moderate cytoplasmic positivity IHC × 200.

Click here for additional data file.

Supplemental Information 18 NF-κB heterodimer expression: Low grade urothelial carcinoma with normal urothelium

(G) Normal urothelium H&E × 200.

Click here for additional data file.

Supplemental Information 19 NF-κB heterodimer expression: Low grade urothelial carcinoma with normal urothelium

(H) Immunostaining with RelA (p65) showed faint to moderate cytoplasmic positivity (Brown colour) IHC × 200. Note: Lymphocytes (in-built control) have taken up stain indicated by arrows.

Click here for additional data file.

Supplemental Information 20 NF-κB heterodimer expression: Low grade urothelial carcinoma with normal urothelium

(I) NF-κB p50 showed moderate cytoplasmic positivity indicated by arrow IHC × 200.

Click here for additional data file.

Supplemental Information 21 RelA Negative control—No primary antibody (Reagent control)

A, RelA subunit staining is a Negative control (Reagent)—No primary antibody was applied on sections

Click here for additional data file.

Supplemental Information 22 p50 Negative control—No primary antibody (Reagent control)

B, p50 subunit staining is a Negative control (Reagent)—No primary antibody was applied on sections

Click here for additional data file.

The authors thank Dr. Caven S. Mcloughlin, Fulbright Specialist, Kent State University, USA for his suggestions. The authors acknowledge the assistance in statistics provided by T Gayathiri, Senior Lecturer, Department of Community Medicine, Sri Ramachandra Medical College and Research Institute (Deemed to be University), India.

Abbreviations

BAD Bcl-2 associated agonist of cell death

Bcl-2 Apoptosis regulator Bcl-2

BCL3 B-cell lymphoma 3 protein

BCR B-cell receptor

BioGRID Biological General Repository for Interaction Datasets

BIRC3/cIAP Baculoviral IAP (inhibitors of apoptosis) repeat-containing protein 3

CBP CAMP Response Element Binding (CREB) - protein

CCL4 C-C motif chemokine 4

CCND1/Cyclin D1 G1/S-specific cyclin-D1

CFLAR/c-FLIP CASP8 and FADD-like apoptosis regulator

CHUK/IKKα Conserved helix-loop-helix ubiquitous kinase/inhibitor of nuclear factor-κB kinase α

CSF-1 Macrophage colony-stimulating factor 1

DIP Database of Interacting Proteins

FFPE Formalin fixed paraffin embedded

GOstats Gene Ontology Statistics

H&E Haematoxylin and Eosin staining

HIFα Hypoxia inducing factor α

HPRD Human Protein Reference Database

HRP Horse raddish peroxidase

ICAM1 Intercellular adhesion molecule 1

IL-8 Interleukin-8

IntAct Molecular interaction database

IκBβ Inhibitor of nuclear factor kappa B β

IKKβ Inhibitor of nuclear factor κB kinase β

MINT Molecular INTeraction database

MIPS-MPact Munich Information Center for Protein Sequences- MIPS Mammalian Protein–Protein Interaction Database

MMP9 Matrix metalloproteinase 9

MYC Myelocytomatosis virus oncogene cellular homolog

NEMO NF-κB Essential Modulator

NF-κB Nuclear factor-kappa B

NF-κB1 Precursor p105 and processed p50

NF-κB2 Precursor p100 and processed p52

NIK NF-κB inducing kinase

PINA4MS Protein Interaction Network Analysis For Multiple Sets

PTGS2/COX2 Prostaglandin G/H synthase 2

pTIS Carcinoma in situ

RelA V-Rel reticuloendotheliosis viral oncogene homolog A

RelB V-Rel reticuloendotheliosis viral oncogene homolog B

SGE Sun Grid Engine

SOD2 Superoxide dismutase 2

TNF Tumor necrosis factor

TP53 Cellular tumor antigen p53

TLRs Toll like receptors

TCR T-cell receptor

TRAF2 TNF receptor-associated factor 2

VCAM Vascular cell adhesion protein

VEGF-A Vascular endothelial growth factor A

XIAP E3 ubiquitin-protein ligase.

Additional Information and Declarations

Competing Interests

Author Contributions

Human Ethics

Data Availability

The authors declare there are no competing interests.

Sankari Durairajan, Mary Divya Samuel, Dicky John Davis G and George Priya Doss C performed the experiments, analyzed the data, contributed reagents/materials/analysis tools, prepared figures and/or tables, authored or reviewed drafts of the paper, approved the final draft.

Charles Emmanuel Jebaraj Walter conceived and designed the experiments, analyzed the data, contributed reagents/materials/analysis tools, authored or reviewed drafts of the paper, approved the final draft.

Dinesh Palani performed the experiments, analyzed the data, prepared figures and/or tables, authored or reviewed drafts of the paper, approved the final draft.

Sneha Pasupati performed the experiments, contributed reagents/materials/analysis tools, prepared figures and/or tables, authored or reviewed drafts of the paper, approved the final draft.

Thanka Johnson conceived and designed the experiments, analyzed the data, contributed reagents/materials/analysis tools, prepared figures and/or tables, authored or reviewed drafts of the paper, approved the final draft.

The following information was supplied relating to ethical approvals (i.e., approving body and any reference numbers):

The Institutional Ethical Committee of Sri Ramachandra Medical College and Research Institute (Deemed to be University), Chennai, India reviewed and accepted the study, approving collection of archival samples deposited at Sri Ramachandra Medical Centre from the year 2008 for this retrospective study (Ref: CSP/14/FEB/33/25).

The following information was supplied regarding data availability:

For Figs. 4 and 5, native photomicrographs for each stain and controls are available as Supplemental Files.

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
