# Peer review of "Differential expression of NF-κB heterodimer RelA/p50 in human urothelial carcinoma"

_PeerJ, doi:10.7717/peerj.5563_

## Round 0.1 · original submission · Major Revisions

You will see that both external reviewers indicate the potential value of your work, but they have also raised many serious problems that will prevent publication at this time.

You should see their comments as being helpful for you to improve your work and I strongly recommend that you respond to each and every comment in a positive fashion.

Their issues range from the methods employed, the reasons for the study, and your interpretation of the data.

Both reviewers have also commented on the style and the language, which will need to be improved to an acceptable level.

Reviewer 1 ·

Basic reporting

In general, this seems like the beginnings of an interesting project identifying P50 and RELA as potential mediators of cancer -specific NFKB signaling in urothelial carcinoma. However, the manuscript is not written in idiomatically correct English, it is disorganized overall, the methods are unclear, and the analysis and statistical statements are often incomplete.

English is uneven. Literature references are satisfactory. Raw data is not shared.

The manuscript needs better organization. For example, the method section for immunohistochemistry does not state controls used for each antibody. In the discussion section, the manuscript explains that staining of macrophages indicates that the two antibodies are working properly. Negative controls are not specified. Both positive and negative controls should be described in the methods section.

A minor criticism: The introduction is a bit over-broad relative to the findings. Much about bladder cancer course and detection is not relevant to the findings or discussion. It can be shortened.

Experimental design

Although not clearly stated, the investigators appear to have used an online tool to examine differential expression of various proteins in benign urothelium vs. cancer. They identified overexpression of NFKB family proteins in cancer relative to benign. Their thresholds and statistical approach were not specified. The samples used in this online research were not identified. Issues such as false discovery correction were not presented. Degree of over-expression was not specified. It is therefore difficult to evaluate this claim.

The authors then go on to evaluate expression of RELA and P50 proteins in a small tissue microarray. In the text that describes the immunohistochemistry and pathologic evaluation, much of the text was very difficult to follow. As an example, the following sentence was uninterpretable: “Histological findings showed invasive high grade UC with focal squamous differentiation (5 %) 234 and tumor free muscle regions at T1 stage (Fig. 4D).”

There is a sentence in the discussion (Line 331) “In our study macrophages served as built-in controls of NF-κB heterodimers staining and indicated anti-tumor response suppression by positively staining for p50 as discussed by others (Saccani et al., 2006; Annunziata et al., 2010; Cartwright et al., 2016).” This raises a question, did the authors find that NF-κB heterodimer expressing macrophages were more abundant in cancer than in benign urothelium?

In the discussion, the authors claim that the results have prognostic significance, but this is difficult to assess, as there is no comparison of event-free or overall survival of patients on the basis of RELA or P50 status.

Validity of the findings

Statistical statements in the text are incomplete: As one example, Line 217, “Comparing different grades in UC by Fisher’s exact test revealed significant associations (Table 4).” Which grades are associated with which disease states? Further, there are claims of associations between cancer stage and levels of protein expression, but these associations are not tested for statistical significance.

Further, the biologic context for the work needs improvement, as it never states whether the findings indicate increased or decreased NFKB signaling in urothelial carcinoma, nor its potential association with stage or grade.

Additional comments

Overall, there is some interesting data that needs refinement and further exploration. Once that is done, the manuscript should be edited by a competent professional for completeness, format, and flow.

Reviewer 2 ·

Basic reporting

Manuscript fails in the following aspects:
- It has English languaje issues
- The Objective is not clear
- There is no hypothesis

I suggest to use English editing services and state clear objective and hypothesis for the work.

Experimental design

In terms of experimental design, the manuscript fails in the following aspect.
-There is not a research question properly defined.
- I recommend to clearly describe how is Cytoscape app used to predict proteins interaction and how results could be interpreted in conjunction with the immunofluorescence assays.

Validity of the findings

See General Comments to the author section.

Some statements in the discussion section are not validated with the results, but could be written as speculations. The following is an example:
The cytoplasmic expression of p50 remained significantly extensive across UC tissues similar to RelA subunit which indicated tumor promotion by the heterodimers in the initial stages.

Additional comments

Based on the provided information, although it is not enough clear in the manuscript, the goal of this work was to evaluate if there is a differential expression of NF-KB subunits RelA and p50 which could be associated with the utorthelial carcinoma (UC) and its aggressiveness.

Although the work seems to be interesting and may be relavant due to the critical role of the transcription factor NF-kb in carcinogenesis, i consider that some important issues need to be clearly evaluated and discussed before considering for publication.

1. Authors may remember that only NF-KB p50 and p52 subunits have the ability to bind DNA but they are not able to activate transcription since they lack the transactivation domain. On the other hand, RelA, RelB and RelC do not have the DNA binding domain, but do have the C-terminus transactivation domain. For this reason, only heterodimers composed of one of the Rel subunits (A, B or C) plus either p50 or p51 are able to activate gene transcription. On the other hand, p50 homodimers are associated with gene repression because they bind to DNA avoiding heterdimers to bind and promote transcription (Marienfeld R et al., 2003).
Accordingly, I suggest to the authors to thoroughly discuss the meaning of the results from immunostaining and the use of cytoscape app, mainly when authors mention that according with cytoscape analysis, p50 expression was inhibited but RelA target genes were over-expressed.
The same request is for what is said in the discussion section, where authors mention that the heterodimer expression revealed disease aggressiveness being directly proportional to RelA nuclear expression but not 50.
Also, I kindly request authors to explain the mechanism through which the cytoplasmic expression of p50 and RelA could induce tumor promotion as is also mentioned in the discussion section, since to activate gene expression they should translocate the nucleus and form heterodimers.


2. Other issues in this manuscript include:
a. In the introduction section I suggest to better justify the study of NF-KB to predict UC status or aggressiveness.
b. The aim of the study is not clear
c. In general, the English language should be reviewed to improve the comprehension of your manuscript. The following are some examples where the language may be improved:

Line 60 “Urothelial carcinoma (UC) in industrialized countries is the fifth most common cancer and second most common genitourinary tract cancer. UC is also called as transitional cell carcinoma (TCC) accounts to 5% of all cancers (Schulz 2006) and are the commonest malignancy of the urinary tract with lesser incidence of the upper urinary tract carcinomas “.

Line 105 “UC is not wholly a resultant of gene mutations as inheritance of genes or mutations leading to UC are rare which is confirmed by microarray expression studies”

Line 269. “Our results revealed suppression of NF-κB1/p50, NF-κB2/p52 and RelB in the stage or grade undefined urothelial tumor tissue presented in PINA4MS app”.

3. There are several typos along the manuscript. For example:
Line 185: were over expressed of in urothelial cancer cells when compared to normal urothelial cells.

---

## Round 0.2 · Major Revisions

Many thanks for your revised version, which has addressed most of the reviewers comments. However, you will see from the reviewer's comments that they are not satisfied with your revised manuscript, despite the many improvements. Specifically, in addition to the extra details of the in silico analysis provided in the revision, I agree that you will need to provide information of the patients (the numbers and clinical details of patients, and the controls) that were used for your in silico analyses. Without these details, it is not possible for a reader to judge the validity (or otherwise) of your claims.
If you can include these details appropriately, there is a good chance that we will accept the article.

Reviewer 1 ·

Basic reporting

Overall improved, although the in silico work is not explained. See section 2.

Experimental design

The manuscript is much improved, but unfortunately does not fulfill the requirements for a mature publication in this field.
The most important problem and the aspect of the paper that is the furthest from taking an acceptable form is the in silico work where no information is given regarding how many patients and samples were analyzed in silico, and what the clinical and pathologic features were for that cohort. Is therefore not possible to review this part of the manuscript for accuracy or for how well the analysis supports conclusions.

Validity of the findings

Beyond the in silico work there is a report of immunohistochemical analysis of two proteins in the NFKB pathway with little solid data as to whether the results indicate increased or decreased activity of the pathway in disease progression. Overall, the revised manuscript does not provide much clarity on its subject matter and it is difficult to see a path for this information towards clinical utility.

---

## Round 0.3 · accepted · Accept

Thank you for responding to the Reviewer's comments and suggestions.

#